# A Genome-Wide Functional Screen Identifies Enhancer and Protective Genes for Amyloid Beta-Peptide Toxicity

**DOI:** 10.3390/ijms24021278

**Published:** 2023-01-09

**Authors:** Pol Picón-Pagès, Mònica Bosch-Morató, Laia Subirana, Francisca Rubio-Moscardó, Biuse Guivernau, Hugo Fanlo-Ucar, Melisa Ece Zeylan, Simge Senyuz, Víctor Herrera-Fernández, Rubén Vicente, José M. Fernández-Fernández, Jordi García-Ojalvo, Attila Gursoy, Ozlem Keskin, Baldomero Oliva, Francesc Posas, Eulàlia de Nadal, Francisco J. Muñoz

**Affiliations:** 1Laboratory of Molecular Physiology, Department of Medicine and Life Sciences, Faculty of Health and Life Sciences, Universitat Pompeu Fabra, 08003 Barcelona, Spain; 2Department of Medicine and Life Sciences, Faculty of Health and Life Sciences, Universitat Pompeu Fabra, 08003 Barcelona, Spain; 3Computational Sciences and Engineering, Koc University, Istanbul 34450, Turkey; 4Laboratory of Dynamical Systems Biology, Department of Medicine and Life Sciences, Faculty of Health and Life Sciences, Universitat Pompeu Fabra, 08003 Barcelona, Spain; 5College of Engineering, Koc University, Istanbul 34450, Turkey; 6Laboratory of Structural Bioinformatics (GRIB), Department of Medicine and Life Sciences, Faculty of Health and Life Sciences, Universitat Pompeu Fabra, 08003 Barcelona, Spain; 7Institute for Research in Biomedicine (IRB Barcelona), The Barcelona Institute of Science and Technology, 08028 Barcelona, Spain

**Keywords:** Alzheimer’s disease, amyloid beta-peptide, genome-wide screening, SURF4, calcium, SOCE

## Abstract

Alzheimer’s disease (AD) is known to be caused by amyloid β-peptide (Aβ) misfolded into β-sheets, but this knowledge has not yet led to treatments to prevent AD. To identify novel molecular players in Aβ toxicity, we carried out a genome-wide screen in *Saccharomyces cerevisiae*, using a library of 5154 gene knock-out strains expressing Aβ_1–42_. We identified 81 mammalian orthologue genes that enhance Aβ_1–42_ toxicity, while 157 were protective. Next, we performed interactome and text-mining studies to increase the number of genes and to identify the main cellular functions affected by Aβ oligomers (oAβ). We found that the most affected cellular functions were calcium regulation, protein translation and mitochondrial activity. We focused on SURF4, a protein that regulates the store-operated calcium channel (SOCE). An in vitro analysis using human neuroblastoma cells showed that *SURF4* silencing induced higher intracellular calcium levels, while its overexpression decreased calcium entry. Furthermore, *SURF4* silencing produced a significant reduction in cell death when cells were challenged with oAβ_1–42_, whereas *SURF4* overexpression induced Aβ_1–42_ cytotoxicity. In summary, we identified new enhancer and protective activities for Aβ toxicity and showed that SURF4 contributes to oAβ_1–42_ neurotoxicity by decreasing SOCE activity.

## 1. Introduction

The aggregation of amyloid β-peptide (Aβ) into oligomers and fibrils is one of the main hallmarks of Alzheimer’s disease (AD) [1,2]. Aβ is generated by the sequential cleavage of the amyloid precursor protein (APP) [3,4,5], which takes place in the secretory pathway. Aβ is released into endosomal compartments and the extracellular space [6,7], from where it can also be re-internalized [8,9,10]. Aβ monomers can aggregate into oligomers, which constitute the most toxic form of the peptide [11,12]. Although we know the enzymes involved in Aβ production from APP and some of the mechanisms utilized by oligomeric Aβ to damage neurons, specific treatments to prevent or to cure the disease are still lacking, beyond acetylcholinesterase inhibitors whose therapeutic efficacy is very limited [13]. It is imperative to identify new targets to treat AD patients, given the high number of patients worldwide and the expected explosion in the disease’s prevalence due to the increase in life expectancy.

Much is known about the toxic effects of Aβ oligomers and fibrils: (i) they produce reactive oxygen species (ROS), such as H_2_O_2_ and OH·, that generate oxidative stress, which oxidizes proteins, lipids and nucleic acids [14] and affects calcium homeostasis due to the damage on calcium ATPases [15]; (ii) they interact with synaptic proteins, mainly NMDAR and alpha-7 nicotinic receptors, impairing their physiological functions [16,17]; (iii) they induce calcium dyshomeostasis, increasing its local concentration by known effects on NMDAR [16] and CALHM1 [18] and unknown mechanisms; (iv) they increase nitric oxide (NO) release due to calcium dyshomeostasis that enhances neuronal NO-synthase activity, allowing the formation of peroxynitrite that nitrates proteins, which mainly leads to the irreversible inactivation of the proteins and even to its misfolding and intracellular aggregation, as it has been proposed for tau protein [19,20]; (v) mitochondrial dysfunction has also been reported in AD as one of the key events in this pathology by decreasing energy supply, increasing oxidative stress, producing NO (by the mitNOS) and collaborating in apoptosis due to the intracellular calcium increase [20,21,22]. In all of these mechanisms, calcium dysregulation appears as the main cause of Aβ toxicity or one of its major downstream effectors.

In the present work, we used *Saccharomyces cerevisiae* to perform a genome-wide screen to identify regulations of amyloid toxicity. The choice of this model system was based on its reduced complexity and on the similarity of most of its molecular signaling pathways to mammals, including those regulating autophagy, apoptosis, mitochondrial function, cellular trafficking and protein homeostasis [23].

## 2. Results

### 2.1. Identification of Enhancer and Protective Genes for Aβ Toxicity in Yeast

We previously demonstrated that Aβ_1–42_ is toxic for *S. cerevisiae* [24]. Other studies have shown that while intracellular Aβ is barely toxic or innocuous to yeast when expressed in the cytosol, it is highly cytotoxic when directed to the secretory pathway [25,26]. These and other studies have thus shown that yeast is a useful model to evaluate intracellular Aβ toxicity [26,27].

To evaluate the toxic effects of Aβ in *S. cerevisiae*, we overexpressed wild-type (WT) human Aβ_1–42_ in those cells, which is the most aggregative and pathogenic isoform of Aβ. Cytosolic expression of Aβ fused to GFP (Aβ-GFP) is known not to induce toxicity [28], whereas Aβ-GFP expressed in the secretory pathway induces cell death [26,27]. The whole procedure is shown schematically in Figure 1 and explained in the Section 4. Interestingly, we fused Aβ to the mating factor α (MFα) pre-pro-leader sequence secretion signal derived from the precursor of the *S. cerevisiae* MFα. In addition, the Aβ construct contains a GFP tag at the C-terminus of the Aβ sequence connected by a linker nucleotide sequence that enables detection of the GFP signal [26] (Figure 2A). Aβ expression was placed under the control of the *GAL1* promoter, so that Aβ could be induced by growing the cells in galactose medium (Appendix A). Two human Aβ_1–42_ mutants termed Dutch (E22Q) and Arctic (E22G) were also cloned and used in this report. Dutch and Arctic Aβ are human genetic mutations very prone to aggregate, and thus induce an early onset of AD [29,30,31,32]. We transformed yeast cells with WT, Dutch and Arctic Aβ_1–42_ constructs, and GFP expression was assessed by Western blot in cells cultured in inducing medium (with galactose) for 6 h at 30 °C (Figure 2B, Appendix A), demonstrating that the expression of the different constructs is homogeneous (Appendix A). Aβ_1–42_ fused to GFP results in a band of ~41 kDa when the MFα is not yet processed (MFα-Aβ_1–42_-GFP), and a band of ~31 kDa when the protein is mature and MFα has been processed (Aβ_1–42_-GFP). In addition, a ~45–50 kDa band is observed, which may represent a glycosylation modification of MFα, according to a previous report [26]. As a negative control, expression of GFP alone results in a band of 26 kDa. Then, we assessed Aβ_1–42_ cellular localization by confocal microscopy images of cells grown in inducing medium for 6 h (Figure 2C) at 30 °C. A cytosolic diffuse pattern is observed in the GFP negative control, whereas in the Aβ_1–42_ containing yeast, a punctuated pattern is observed, probably corresponding to endomembranous localization of aggregated amyloid as we also obtained this by ultrastructural analysis (Appendix A).

Since the main objective of this study is to investigate the effect of Aβ_1–42_ on cell survival, we analyzed yeast growth in the presence of the peptide (Figure 2D). Quantification of the growth rate was calculated as the growth detected after 3 days in solid plates with inducing medium, divided by the growth in non-inducing medium (Gal/Glu) (Figure 2E). We found that the induction of WT Aβ_1–42_ expression strongly impaired yeast growth compared to the non-inducing medium condition, consistent with the expected toxic effect of WT Aβ_1–42_. Such a reduction in growth was not observed when comparing yeast transformed with an empty vector (control) in inducing medium. In addition, we observed an increased cytotoxicity induced by both Dutch and Arctic Aβ_1–42_ compared to WT Aβ_1–42_, suggesting that Aβ_1–42_ toxicity depends, at least partially, on the aggregative proneness of the peptide, in agreement with previous findings [26]. These data confirm that *S. cerevisiae* containing the described Aβ_1–42_ construct is an appropriate model to evaluate Aβ toxicity.

The peptide used in the present study to screen for Aβ toxicity modulators is the WT Aβ_1–42_, from now on referred to simply as Aβ. Modifiers of Aβ toxicity were identified by screening 5154 mutants from a *S. cerevisiae* genome-wide deletion library, which were mated with an Aβ-containing strain, and strains containing the mutation and the inducible Aβ construct were selected using an automated system [33] following the protocol described in the Section 4. The wild-type (WT) strain, which does not contain Aβ (empty plasmid), showed a growth rate of 1.04 after 4 days and 1.83 after 5 days of galactose induction, whereas the WT strain containing Aβ presented a growth rate of only 0.09 after 4 days and 0.39 after 5 days of induction (Appendix A). Mutant strains with a minimum growth rate of 0.7 after 4 days of induction were considered protectives regarding Aβ and knock-out strains with a growth rate of 0 after 5 days of induction were considered enhancers of Aβ toxicity (Appendix A). Knock-out strains for genes with reported deficient growth in galactose medium and strains with observed deficient growth in glucose medium were discarded. Finally, we performed an analysis with a volcano plot (Appendix A) to identify the enhancer genes regarding amyloid toxicity. Taking into consideration these conditions, we identified 141 strains that contribute to Aβ toxicity since their deletion causes a decreased toxicity (Appendix A). Furthermore, 312 strains were considered protective against Aβ since their deletion shows an increase in Aβ toxicity (Appendix A).

The pathophysiological relevance of these findings is supported by the fact that among the regulatory genes for amyloid toxicity identified, there are 10 genes that have been proposed in GWAS as playing an important role in AD: *AQP9* [34,35], *HSPA1B* [34,36], *HSPA1L* [34,37], *OSBPL10* [34,38], *AP3M2* [34,39], *GAS7* [34,40], *KMO* [34,41], *RAB5A* [34,42], *UBE4A* [34,43] and *VSP28* [34].

### 2.2. Interactome Analysis

We next performed an interactome analysis to identify the intracellular pathways involved in toxicity and protection. Two main approaches were used for this analysis. One of them was to perform pathway enrichment for the mammalian genes, and the second one was to analyze the network components in the protein–protein interaction network constructed by the mammalian genes.

When reactome pathway enrichment was performed on the 238 mammalian orthologues, mainly mitochondrial-related pathways were found with a significant False Discovery Rate (FDR) and *p*-value (Figure 3).

Out of the 238 mammalian orthologues identified, 81 of these were enhancer genes of Aβ toxicity (Appendix A) and 157 were protective genes (Appendix A). The amyloid toxicity protective or activator genes were arranged into a network using StringDB. To that end, we utilized the 238 mammalian orthologue genes as input, and StringDB returned 188 of the mammalian genes in *Homo sapiens*. Appendix A demonstrates the proteins interacting with each other. Then, we used this network to identify genes that seemed to be crucial in the network architecture. Specifically, Appendix A shows the genes with the highest average shortest path length (closeness to the other proteins), highest betweenness centrality and highest degree (hub). Data from Appendix A show that the majority of protective genes are hub genes (Figure 3); moreover, these genes are mainly mitochondrial ribosomal proteins. Furthermore, the properties of the genes with the highest betweenness centrality are distributed equally as protective and enhancer.

Furthermore, we identified densely interconnected nodes (modules) in the network. Figure 4 shows the module with the highest Molecular Complex Detection (MCODE) scores in yellow.

We performed pathway enrichment of the genes highlighted in yellow and we observed that mitochondrial translation-related pathways have an FDR less than 0.05. Specifically, mitochondrial translation elongation, mitochondrial translation initiation, mitochondrial translation termination, mitochondrial translation and translation are enriched, respectively, according to their FDR values (Appendix A).

Next, we carried out a text-mining analysis of the reported genes (Appendix A), finding the potential role of the store-operated calcium channel (SOCE). To further examine the potential connection between SURF4 and the pathways involved in SOCE, we used GUILDify v2.0, text mining Uniprot and DisGenet datasets, with the genes *SURF4*, *STIM1*, *ORAI1* and *CRACR2*. The search resulted in several genes associated with them, including pathways of ubiquitination, *KIF5A*, *SARAF* and *SGK1*. Then, we used GUILDify v2.0 and the human interactome to extend and analyse the subnetwork linking these genes (Figure 5). The analysis also highlighted some other genes, such as *COLEC12*, that appeared in the set of genes associated with AD (DisGenet_Guild).

### 2.3. Modulators of Aβ Toxicity Associated with Ca^2+^ Homeostasis

SURF4 has been proposed to modulate store operating calcium entry (SOCE), an important mechanism that controls cellular calcium homeostasis [44]. To elucidate the effect of SURF4 in SOCE, we transfected human neuroblastoma cells with either a selective siRNA for silencing *SURF4* (Figure 6A) or a plasmid to express SURF4 (Figure 6D), which resulted in a significant decrease (Figure 6A) or increase in mRNA SURF4 expression (Figure 6D), respectively. Then, we carried out viability assays in order to study whether SURF4 expression affected Aβ toxicity in mammalian cells. We found a lower Aβ toxicity when human neuroblastoma cells were transfected with *SURF4* siRNA (Figure 6B,C) and an increase in cell toxicity when SURF4 was overexpressed (Figure 6E,F).

We tested the role of SURF4 expression on basal intracellular calcium. SHSY5Y cells depict a tendency to increase basal calcium levels when *SURF4* is silenced (Figure 7A). The effect is the opposite after SURF4 overexpression, and a significant decrease in basal calcium is observed (Figure 7D). These results are most probably related to the effect of SURF4 on Ca^2+^ entrance from the extracellular medium. Thus, when evaluating SOCE following ER Ca^2+^ depletion by thapsigargin, we only observed significant differences in intracellular Ca^2+^ changes due to altered SOCE upon addition of Ca^2+^ to the extracellular medium, which was enhanced or impaired by *SURF4* silencing or overexpression, respectively (Figure 7B–F).

According to the reported effect of SURF4 as a repressor of SOCE activity and in agreement with the results obtained after SURF4 overexpression, the direct inhibition of STIM-ORAI interaction by Ro also increases Aβ toxicity (Figure 8A,B).

## 3. Discussion

In a previous work, we demonstrated that exogenous Aβ_1–42_ aggregates are toxic for the yeast *S. cerevisiae* [24]. In that work, we added the aggregated Aβ_1–42_ in the cell culture medium, as it is a common procedure with neuronal cells, and we obtained that extracellular amyloid is toxic for yeasts. The mechanisms involved in the amyloid toxicity were not studied on that occasion, but we hypothesized they implied damages by oxidative stress [14] and unknown mechanisms due to the direct effect on the cell membrane and intracellular effects due to the re-uptake of the aggregates by the yeast cells. Here we studied the effect of endogenously produced Aβ_1–42_. We overexpressed the Aβ_1–42_ to be included in the secretory pathway and re-uptaken by the cells mimicking the Aβ_1–42_ trafficking that occurs in neurons. The fact that some works report non-significant toxicity when Aβ_1–42_ is overexpressed but not included in the secretory pathway could be related with the Aβ_1–42_ inclusion in non-harmful packages, low levels of expression, a rapid degradation by the proteosome or unspecific enzymes, or even the lack of interaction with key compartments or molecules inside the cell.

The present report firstly confirms that Aβ overexpression in the yeast secretory pathway is an appropriate model for identifying modifiers of intracellular Aβ toxicity among the whole yeast genome. Our results suggest that these genes, whose deletion reverts Aβ toxicity, probably contribute to its toxicity since when they are knocked out, Aβ does not exert its cytotoxic effects as it does in controls. On the other hand, genes whose deletion enhances Aβ toxicity can be identified as protectives from Aβ toxicity since when they are knocked out, Aβ toxicity is increased. Alternatively, the effect of these genes can be interpreted as belonging to critical molecular pathways responsible for Aβ toxicity, whose absence would cause an increased vulnerability by some other proteins from the same cellular pathways.

The main aim of constructing the protein–protein interaction network with the protective and activator genes was to analyze the interactions between the given proteins. This network reveals that the toxicity protecting genes have greater links with one another, whereas activators are not as well connected within themselves. Moreover, all of the “Vacuolar protein sorting-associated proteins (VPS)” play a protective role. *VPS18* and *VPS33A* have the highest average shortest path length together with *SURF4*. This indicates that these genes are not central to the network, and specific pathways must be pursued to reach those nodes in this network. Mitochondrial ribosomal proteins are hubs for the network and can play a central role in the crosstalk between protectives and enhancers.

Modules in biological networks are subnetworks within the whole network that are highly interconnected [45]. The detection and analysis of modules in networks, the protein–protein interaction networks in the present work, are thus crucial to apprehend the biological meaning of the network [46]. Therefore, we detected and analyzed the most highly interconnected module on the network. In this module (Figure 4), while RPS11 is an Aβ toxicity activator protein, the rest of the proteins in the module are Aβ toxicity protective proteins. This could mean that since protective proteins have so many connections within themselves, they are unlikely to accommodate mutations [47]. Furthermore, this interconnected module is primarily composed of mitochondrial ribosomal proteins.

Reactome pathway enrichment of all the 238 mammalian orthologues and the module demonstrated that mitochondrial translation is the major significant pathway related to these genes. The mitochondrial translation is a term that encompasses initiation, elongation and termination [48]. All these pathways were enriched in our results. Moreover, malfunctions in mitochondrial translation were shown to be related to many diseases, including neurodegenerative diseases [48,49]. In mammals, mitochondrial translation occurs in mitoribosomes, and these ribosomes produce proteins that play a part in oxidative phosphorylation [48,50,51]. As previously pointed out, there is a relation between SOCE and calcium regulation in mitochondria [52,53,54,55]. Our pathway enrichment results also support this hypothesis. Both mitochondrial production of ROS and calcium are known to regulate the signaling between mitochondria and ER [56,57]. Although the exact mechanism is still unknown, we can hypothesize that disruptions in mitochondrial translation pathways play a crucial role in ROS production and calcium communication between ER and mitochondria.

On the other hand, there are multiple proteins for which their function in the cell is poorly understood. This is the case of SURF4, a housekeeping protein of 37 kDa present in the endoplasmic reticulum and proposed previously to regulate SOCE [44]. Calcium is one of the most important intracellular messengers, contributing to multiple mechanisms that provide a tight control of cytosolic calcium, mainly by calcium-binding proteins such as calmodulin and calbindin and by regulators of the endoplasmic reticulum, the major calcium store. SOCE is the main mechanism to replenish the endoplasmic reticulum. STIM1 is as a calcium sensor that dimerizes when there is a calcium depletion STIM1 and binds ORAI1, a calcium channel, in order to promote extracellular calcium entry. In the present study we found a decisive modulation in SOCE by both the overexpression and interference of SURF4, which produce either a decrease or an increase in calcium entry, respectively. Moreover, basal intracellular calcium concentration is affected in a similar manner by changes in SURF4 expression, which further support the importance of SOCE in calcium regulation.

Calcium is known to be relevant for cellular function, especially in neurons due to its role in neurotransmission. In the present paper, we highlight the importance of SURF4 as a regulator of intracellular calcium by identifying it as a novel player in Aβ-mediated cell death. A higher expression of SURF4 contributes to higher Aβ neurotoxicity related with a decrease in calcium entrance by SOCE. Therefore, we propose that SOCE contributes to neuronal protection against Aβ, as demonstrated when we silenced SURF4.

In summary, we found a set of neuroprotective proteins against Aβ toxicity, and a complementary set of proteins that contribute to Aβ toxicity, such as SURF4. The maintenance of SOCE integrity seems to be crucial to protect neurons against Aβ toxicity. Further work is needed to elucidate the role of the other proteins found in this study regarding their capability to modulate the toxic effect of Aβ in neurons.

## 4. Materials and Methods

### 4.1. Yeast Media and Growth Conditions

Yeast strains were grown in YPD medium (2% *w*/*v* dextrose, 2% *w*/*v* peptone, 1% yeast extract), or synthetic minimal medium (SD; 2% *w*/*v* glucose, 0.17% yeast nitrogen base without amino acids, 5% *w*/*v* ammonium sulfate, 0.003% adenine). Solid medium contains also 20 mg/L agar. When indicated, SD was supplemented with 0.7 g dropout medium without uracil (SD-URA). The inducing medium is SD-URA that contains 2% *w*/*v* galactose instead of glucose as carbon source. Since ammonium sulfate inhibits the function of the antibiotic geneticin (G418), synthetic medium G418 was made with monosodium glutamate (MSG) as a nitrogen source as follows: 20 mg/L agar, 1.7 g/L yeast nitrogen base w/o ammonium sulphate and amino acids, 1 g/L monosodium glutamic acid, 2 g/L amino acid dropout and 2% glucose. Medium preparation and yeast culturing was carried out according to standard techniques.

### 4.2. Yeast Strains and Plasmids

Human Aβ_1–42_ (Aβ42) was overexpressed in yeast as follows. S. cerevisiae strain BY5563α was transformed with a multicopy yeast-expression plasmid (pRS426) with the URA3 selectable marker and *CYC1-GAL1* promoter controlling the expression of MFα-Aβ42-GFP (kindly gifted by C. Marchal, Université Bordeaux 2, France), which was completely sequenced (Appendix A). The construct contains a BamHI restriction site followed by the α-factor prepro sequence, the Aβ_1–42_ coding sequence, the linker GGTGCTGGCGCCGGTGCT and the GFP sequence followed by a Bsu36I restriction site. Plasmid transformations were performed using the lithium acetate method [30]. Transformants were selected in solid synthetic minimal medium without uracil (SD-URA). Colonies were grown for 4 days at 30 °C. Expression of Aβ was induced by growth-in inducing medium (containing 2% *w*/*v* galactose) instead of non-inducing medium (containing 2% *w*/*v* glucose) for at least 5 h at 30 °C.

### 4.3. Yeast Systematic Genetic Screen

The library used for the screen is constituted by 5154 homozygous knock-outs (Saccharomyces Gene Deletion Project, EUROSCARF, Open Biosystems, Huntsville, Alabama, USA). These knocked out cells were produced in the BY4741 strain background (MATa his3Δ1 leu2Δ0 met15Δ0 ura3Δ0) and present geneticin (G418) resistance. Solutions of canavanine (L-canavanine sulfate salt; Can) and G418 were previously dissolved in water at 100 mg/L, filtered sterilized and stored in aliquots at 4 °C. The screen was performed with an automated system using the ROTOR HDA Singer Instruments (Roadwater, Watchet, United Kingdom) following the Synthetic Genetic Array (SGA) protocol with some modifications [33,58]. Briefly, MATα Aβ42-expressing strain (BY5563) was grown overnight (o.n.) in rich medium and transferred to a 96-well plate. Then the cells were pinned onto YPD plates and grown for one day. After pinning the query strain onto fresh YPD plates, the 5154 MATa library strains (BY4741) were pinned on top of the query strain and plates were incubated for one day at 30 °C to allow the mating. Heterozygous MATa/α diploid cells were pinned onto SD with 200 mg/L G418 lacking uracil (SD-URA G418). After a recovery in YPD medium, sporulation was induced for 7 days at 22 °C in medium with low amount of nutrients (2% agar, 1% potassium acetate, 25% Drop out –URA supplemented with uracil). The MATa spore progeny was selected by pinning the spores onto specific selectable medium and grown for 2 days. Specifically, three selection rounds were performed with MSG and the specific markers. First selection medium was MSG–His–Arg Can (50 mg/L) G418 (200 mg/L). Second selection medium was MSG –His –Arg –URA Can (50 mg/L) and third selection medium was MSG–His–Arg–URA Can (50 mg/L) G418 (200 mg/L). The resulting haploid cells containing both the deleted gene and the Aβ plasmid were pinned onto plates with synthetic minimal medium lacking uracil (SD-URA) plates. Cells were placed in either glucose (uninduced) or galactose (induced) medium and incubated at 30 °C for 3, 4 and 5 days before scoring (Appendix A). The screen was performed in duplicate. Analysis of the dots was accomplished with Cell Profiler software (v.4.2.5) and the growth rate was quantified through the ratio of the growth observed in the induced plate after 3, 4 and 5 days over the growth observed in the uninduced plate after 3 days (Gal/Glu). Yeast strains considered enhancers and revertants of Aβ toxicity were analyzed with BiNGO plugin using Cytoscape software (v.3.8.2.). Human orthologues were obtained from Ensembl (http://www.ensembl.org (accessed on 7 October 2021)) and from Drosophila RNAi Screening DRSC Integrative Ortholog Prediction Tool (http://www.flyrnai.org/cgi-bin/DRSC_orthologs.pl (accessed on 7 October 2021)). Further analyses were performed in the Saccharomyces Genome Database (http://www.yeastgenome.org/ (accessed on 7 October 2021)).

### 4.4. Western Blot in Yeast

Yeast cells were harvested and lysed with TCA 85% for 10 min at room temperature (RT). Supernatants were resolved in SDS-PAGE. Gels were transferred in polyvinylidene fluoride membranes (ImmobilonP, Millipore, Burlington, MA, USA), which were blocked for 1 h in tween-tris buffer saline (TTBS) plus 5% milk or 3% bovine serum albumin (BSA). Membranes were incubated o.n. at 4 °C with the following primary antibodies (Abs): 1:500 anti-GFP (Sigma, San Luis, MI, USA) and 1:500 anti-Aβ 6E10 (Covance, Princeton, NJ, USA). Membranes were washed thrice with TTBS and incubated for 1 h with 1:2000 anti-mouse secondary Abs (GE-Healthcare, Chicago, IL, USA). Three washes with TTBS were performed and membranes were developed with Super signal West Pico and Femto Chemiluminiscent substrate (Thermo Scientific, Waltham, MA, USA). Blotting quantification was performed with Quantity One software software (v.4.6.8).

### 4.5. Confocal Microscopy in Yeast

Yeast strains were cultured in inducing medium 6 or 15 h at 30 °C. Then, they were incubated 30 min in poly-lysinated coverslips and digital images were taken with a Leica TCS SP5 II CW-STED confocal microscope, deconvolved with Huygens (SVI) and analyzed with Image J software (v.1.53c).

### 4.6. Yeast Spotting Assays

Strains were grown o.n. at 30 °C in SD-URA containing glucose. Cell concentrations (OD660) were adjusted at OD660 0.3 and after 5 h growing in SD-URA containing raffinose, three dilutions 1:10 were spotted in plates with SD-URA containing glucose (uninduced) or galactose (induced). Plates were incubated at 30 °C for 3 days before analysis.

### 4.7. Yeast Revalidation Assay

The selected knock-out strains were grown in YPD G418 solid medium at 30 °C for 24–48 h plates. One colony of each strain was grown in YPD liquid medium at 30 °C o.n. Cell concentrations were adjusted at OD660 0.2 and after 4 h of growing in YPD liquid medium at 30 °C, cells were transformed with the Aβ construct or an empty vector pRS426 as previously described. Transformants were selected in SD-URA solid medium for 4 days at 30 °C. Then, two colonies of each knock-out strain were selected and used for spotting assay using the ROTOR HDA Singer Instruments as previously described. Briefly, knock-out strains transformed with Aβ construct or empty vector were grown in SD-URA liquid medium o.n. at 30 °C. Cell concentrations were adjusted at OD660 0.2 and after 4 h of growing in SD-URA, cells were pinned into either glucose (uninduced) or galactose (induced) medium and incubated at 30 °C for 2 to 4 days before scoring.

### 4.8. Yeast Growth Curve

Knock-out strains transformed with Aβ construct or empty vector were grown in SD-URA solid medium o.n. at 30 °C. Cells were pinned into SD-URA liquid medium containing either glucose or galactose and incubated at 30 °C for 48 h in a fluorimeter while OD660 values were being registered. OD660 results were used to obtain the growth curves of each mutant.

### 4.9. Interactome Analysis

A network of the amyloid toxicity protective or activator genes was created by using StringDB [59]. The 238 mammalian orthologue genes were utilized as input, and StringDB found 188 of the mammalian genes in *Homo sapiens*. While creating the network, “experiments”, “databases” and “gene fusion” were used as sources. The confidence was chosen to be medium (0.4), and no additional interactors were chosen.

Cytoscape’s [60] network analysis tool was used to find genes that appeared to be crucial in the network and visualization. In particular, the shortest path length, betweenness centrality and degree for all of the network were calculated. We utilized MCODE [61] in order to detect the modules on the network with the following settings: 0.2 for Node Score cutoff, 2 for KCore and 100 for Max. Depth. We used Webgestalt [62] for all the pathway enrichment analysis.

### 4.10. Data Processing and Text-Mining

Text search of the yeast genes was performed in Uniprot and DidGeNet to obtain related proteins published in the Net (Uniprot), thus increasing the number of relevant proteins/genes. We generated a list of AD genes reported in humans in the Net and we performed a text-mining search of this list using GUILDify v2.0 Web Server with DisGeNet data (DisGenet_Guild). Finally, we studied the overlap of the protective and activators genes with the list DisGenet_Guild using the GUILDify v2.0 Web Server.

### 4.11. Human Cell Line Culture

Human neuroblastoma cells (SH-SY5Y cells) were grown with Ham’s F12 GlutaMax (F12 medium; Gibco, Billings, Massachusetts, United States) supplemented with 15% fetal bovine serum (FBS; Gibco) and 1% penicillin/streptomycin (Gibco). Cells were incubated at 37 °C in a humidified atmosphere containing 5% CO_2_.

### 4.12. SURF4 Silencing in Neuroblastoma Cells

SH-SY5Y cells at ~80% of confluence were cotransfected with *SURF4* siRNA from Ambion or a negative siRNA control from Qiagen (Hilden, Germany) with a pGFP reporter. RNAiMax transfection kit (Thermo Fisher technology, Waltham, MA, USA) was used to transfect cells in Opti-MEM. A total of 180 pM siRNA were added to cells plus 0.6 μg of pGFP reporter plasmid for cotransfection in 6-well plate and 72 pM siRNA plus 0.24 μg of pGFP in 24-well plate. Cells were transfected and incubated with growth media for 48–72 h.

### 4.13. SURF4 Overexpression in Neuroblastoma Cells

pCMV-SPORT6-SURF4 (SURF4) plasmid was obtained from BioCat (Heidelberg, Germany). SURF4 gene was recombined into a *pCDNA3.1* YFP tagged plasmid using a sequence with EcoR1 and Xho1 as restriction sites (Appendix A). Then, 80% confluent SH-SY5Y cells were transfected using 3 μg (6-well plate) or 1 μg (24-well plate) of plasmidial DNA. For GFP cotransfection, the ratio between the plasmid and the reporter was established as 1:10 to ensure the presence of the plasmid in the GFP positive cells. Lipofectamine 3000 transfection kit (Thermo Fisher Scientific) was used to transfect according to distributor’s manual, using Opti-MEM (Gibco) as solution media. Cells were transfected and incubated with growth media for 48–72 h.

### 4.14. Aβ_1–42_ Oligomers (oAβ_1–40_) Preparation

A total of 1 mg lyophilized Aβ_1–42_ wild-type (Anaspec, Fremont, CA, USA) was solubilized in 250 µL of MilliQ water. The pH was adjusted to ≥10.5 with 1 M NaOH to avoid the isoelectric point of Aβ. A total of 250 µL of 20 mM phosphate buffer (pH 7.4) was added to neutralize pH and samples were sonicated for 1 min in a bath-type sonicator (Bioruptor, Diagenode, Liege; Belgium). Aliquots were prepared and dissolved to 0.4 mg/mL (88.6 µM) in serum-free Ham’s F12 GlutaMax (F12 medium; Gibco) to treat cells. Aβ_1–42_ was incubated for 24 h at 4 °C to allow its oligomerization [15]. Aliquots of oAβ_1–40_ were dissolved to 0.4 mg/mL (88.6 µM) in serum-free F12 medium to treat cells.

### 4.15. Neuroblastoma Cell Viability Studies

SH-SY5Y cells were seeded in 24-well plates (7.5 × 10^4^ cells/well). After 12 h, the growth medium was removed. Cells were treated with 5 or 10 µM oAβ_1–42_ in F12 medium without FBS for 24 h at 37 °C. In one set of experiments, we pretreated for 1 h with 500 nM RO2959 hydrochloride (Ro) to inhibit STIM/ORAI interactions. Cell survival was assayed by 3-(4,5-dimethylthiazol-2-yl)-2,5-diphenyltetrazolium bromide (MTT) reduction method. We added 10% MTT stock solution (5 mg/mL) for 2 h. Then, medium was discarded and 100 μL of DMSO was placed per well. Absorbances were measured in a plate reader (BioRad, Hercules, CA, USA) at 540 nm and 650 nm (as reference). Control cells were assumed as 100%.

### 4.16. Quantitative PCR Measurement for Transcriptional Studies in Neuroblastoma Cells

SH-SY5Y cells were seeded in 6-well plates (30 × 10^4^ cells/well) to growth up to ~80% confluence. Then, cells were transfected to overexpress or to silence *SURF4* as indicated above. Finally, the mRNA of SH-SY5Y cells was extracted by using the Total RNA Isolation kit (NZYtech, Lisbon, Protugal) and quantified with NanoDrop ND-1000 (Thermo Fisher Scientific). Total cDNA was obtained from 500 μg of mRNA by retrotranscription using SuperScript III Reverse Transcriptase kit (Invitrogen, Waltham, MA, USA). Quantitative PCR was performed for *SURF4, GAPDH and HPRT* by using the fluorophore Sybr Green (Thermo Fisher Scientific) along with 10 μM cDNAs specific primers. The experiment was performed with QuantStudio 12K Flex Real-Time PCR System (Thermo Fisher Scientific). ΔΔCT was measured using GAPDH or HPRT as housekeeping genes.

### 4.17. Basal Calcium Measurements in Neuroblastoma Cells

SH-SY5Y cells were seeded in coverslips (3 × 10^4^ cells/well in 24-well plates) to growth up to ~80% confluence. Then, cells were transfected to overexpress or to silence *SURF4* as indicated above. Cells were treated with 5 μM Fura 2AM (Invitrogen) supplemented with 0.02% pluronic acid for 40 min in an isotonic solution (1.2 mM CaCl_2_, 2.5 mM KCl, 0.5 mM MgCl_2_, 140 mM NaCl, 5 mM glucose and 10 mM HEPES; 305 mOsm; pH 7.4). Experiments were performed at RT using a custom-made chamber. Fluorescence images were obtained using a Nikon inverted microscope with a xenon lamp. GFP positive transfected cells were selected using 450 nm excitation and a camera managed by AquaCosmos software (v.1.3. Hamamatsu Photonics, Hamamatsu, Japan). Cytosolic calcium levels were obtained using 340/380 ratio. Measurement of basal calcium was obtained along 4 min recording each 5 s.

### 4.18. Endoplasmatic Reticulum Calcium Release in Neuroblastoma Cells

After basal calcium measurement, cells were bathed in a free calcium solution (2.5 mM KCl, 1.7 mM MgCl_2_, 140 mM NaCl, 5 mM glucose, 0.5 mM EGTA and 10 mM HEPES; 305 mOsm; pH 7.4). Endoplasmic reticulum (ER) calcium release was measured using Fura 2AM. Positive controls with 1 μM thapsigargin, an ER calcium depletion drug, was carried out in free calcium solution until peak recovery.

### 4.19. Study of Store Operated Calcium Channel in Neuroblastoma Cells

Store-operated calcium channel (SOCE) was measured after ER calcium release by changing the extracellular free calcium solution with a solution containing 1.2 mM Ca_2_Cl. Both solutions were supplemented with 1 μM thapsigargin to avoid ER calcium interference.

### 4.20. Statistical Analysis

Data are expressed as mean ± SEM of n experiments as indicated in the corresponding figures. Statistical analyses for qPCR were performed by two-way Student t-test or two-way ANOVA plus Bonferroni as post hoc test. The software used was GraphPad software. For calcium, a Student’s t-test was performed comparing means for basal calcium, and areas under the curve for reticular calcium and SOCE.

## Figures and Tables

**Figure 1 ijms-24-01278-f001:**
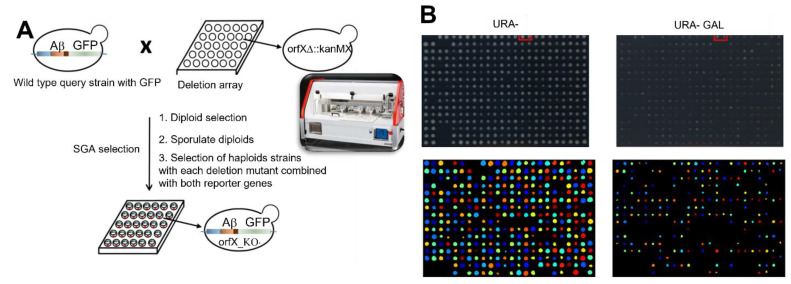
Genetic screening for modulators of Aβ toxicity. (**A**) Mating GAL10-MFα-Aβ_1–42_-GFP strain with KO yeast collection. A yeast strain overexpressing MFα-Aβ_1–42_-GFP was mated with a yeast collection of 5154 mutants. A synthetic genetic array (SGA) was performed as indicated in the M&M section to obtain mutant KO yeasts that overexpress MFα-Aβ_1–42_-GFP. (**B**) Phenotype comparison with and w/o galactose (upper panels) and quantification with Cell Profile (lower panels).

**Figure 2 ijms-24-01278-f002:**
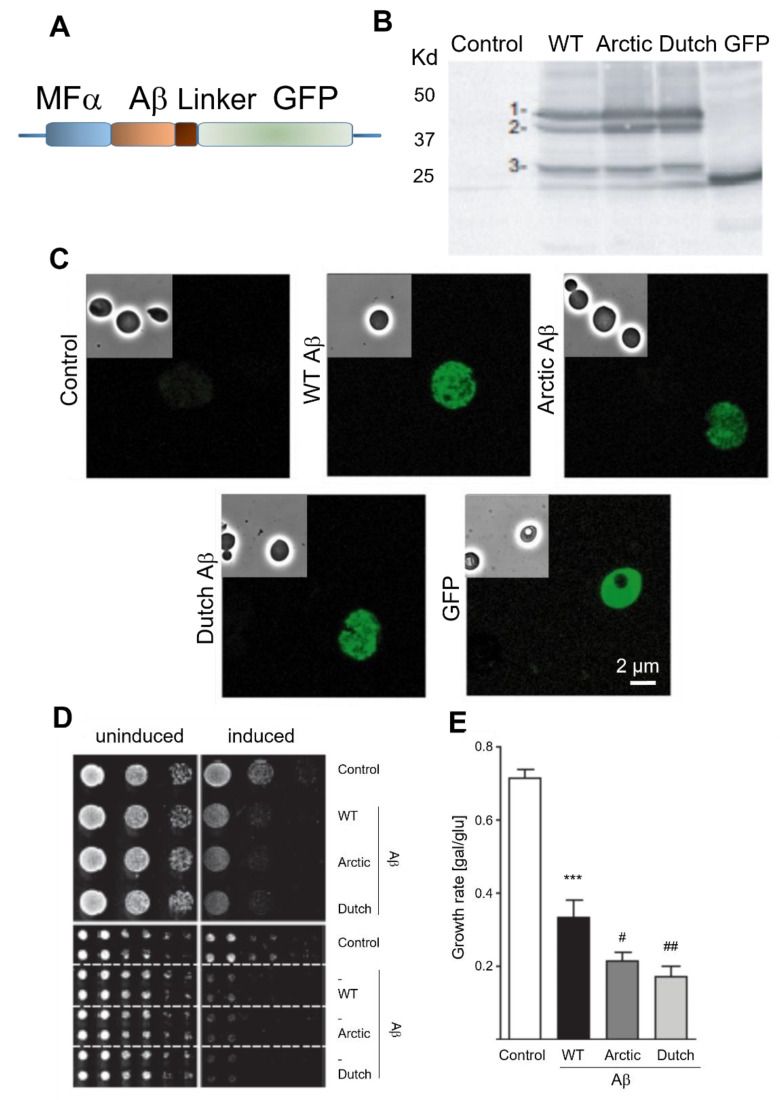
Expression of Aβ_1–42_ and its effect in yeast. (**A**) Aβ_1–42_ construct contains the mating factor α (MFα) pre-pro-leader sequence secretion signal at the N-terminal and a GFP tag at the C-terminal fused with a gly-ala linker (in brown). (**B**,**C**) Western blot analysis of Aβ_1–42_-GFP expression using an anti-GFP antibody (**B**) and representative Aβ_1–42_-GFP confocal images (**C**) in yeast transformed with wild-type (WT), Dutch or Arctic Aβ_1–42_ or with an empty vector (control) and cultured for 6 h at 30 °C in inducing (galactose) medium. A strain constitutively expressing GFP was used as a positive control. Glycosylated MFα-Aβ_1–42_-GFP is labelled as 1, non-glycosylated MFα-Aβ_1–42_-GFP is marked as 2 and Aβ_1–42_-GFP corresponds to 3. (**D**) Serial dilutions of yeast transformed with WT, Dutch or Arctic Aβ_1–42_ or with an empty vector (control) and spotted on inducing (galactose) and non-inducing (glucose) medium for 3 days at 30 °C. (**E**) Quantification of mean growth calculated after 3 days in inducing medium divided by the growth in non-inducing medium (Gal/Glu). Data are the mean ± SEM of 11–16 experiments. *** *p* < 0.001 vs. control, ^##^ *p* < 0.01, ^#^ *p* < 0.05 vs. WT by ANOVA plus Bonferroni as post hoc test.

**Figure 3 ijms-24-01278-f003:**
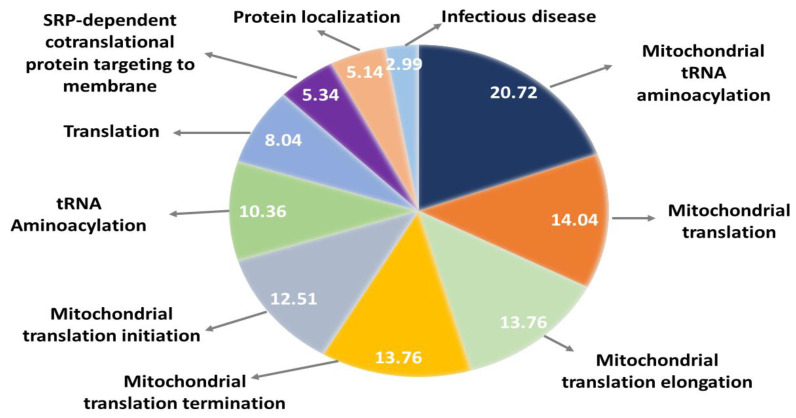
Reactome pathway enrichment of the network created with the 238 mammalian orthologues. Enrichment ratios are given for each pathway. Enrichment ratios were obtained from WebGestalt.

**Figure 4 ijms-24-01278-f004:**
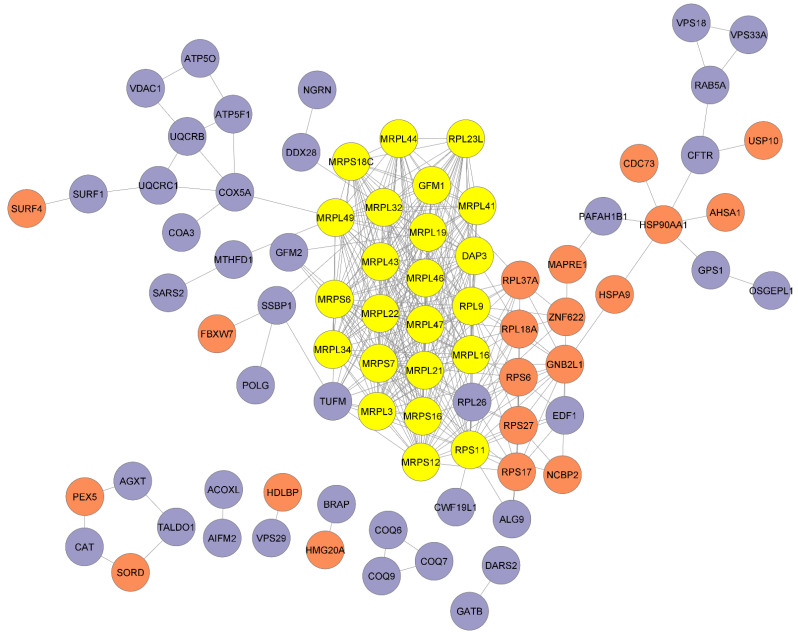
Cluster with the highest MCODE score is shown in yellow. Amyloid toxicity protective genes can be seen in blue and enhancer genes can be seen in salmon.

**Figure 5 ijms-24-01278-f005:**
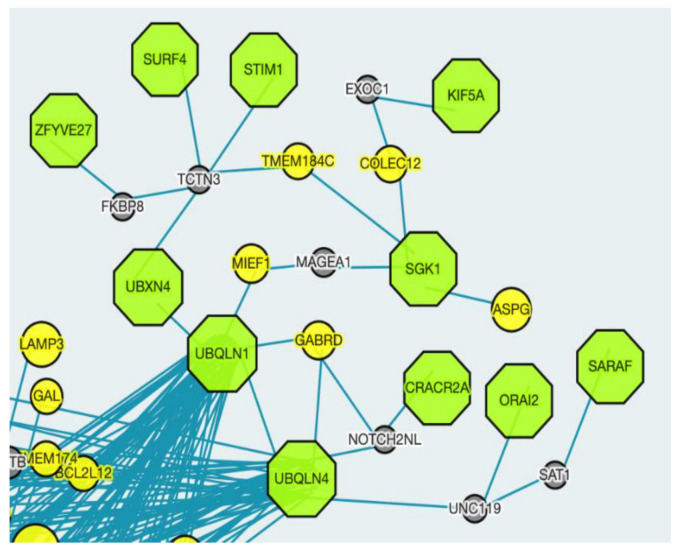
Subnetwork of genes associated with SOCE and connected to *SURF4*.

**Figure 6 ijms-24-01278-f006:**
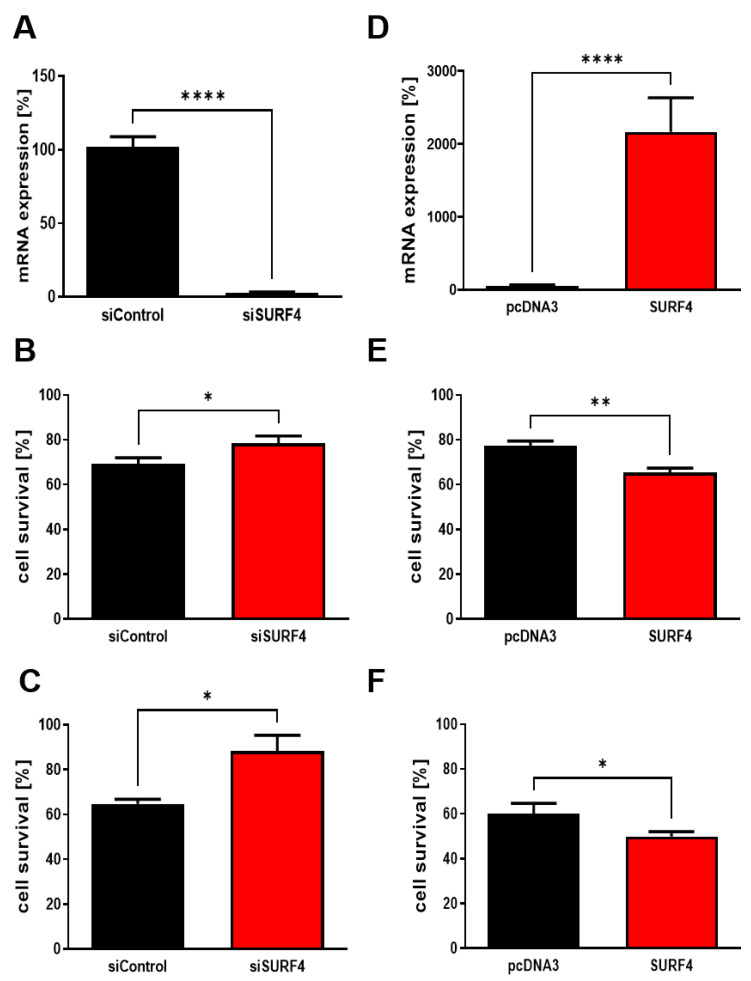
SURF4 contributes to Aβ_1–42_ toxicity on neuroblastoma cells. (**A**) Human neuroblastoma cells were transfected with *SURF4* siRNA or with a non-active control siRNA, and after 48 h the levels of *SURF* mRNA were quantified by semi-quantitative rtPCR. Data are the mean ± SEM of 3 independent experiments. **** *p* < 0.0001 vs. control by Student’s t-test. (**B**,**C**) Cells transfected with *SURF4* siRNA were treated with 5 µM (**B**) or 10 µM (**C**) oAβ_1–42_ for 24 h. Data are the mean ± SEM of 3–10 independent experiments. * *p* < 0.05 vs. control by Student’s t-test. (**D**) Human neuroblastoma cells were transfected with a plasmid containing the sequence of human *SURF4* or with a non-coding control (*pcDNA3*), and after 48 h the levels of *SURF* mRNA were quantified by semi-quantitative rtPCR. Data are the mean ± SEM of 3 independent experiments. **** *p* < 0.0001 vs. control by Student’s t-test. (**E**,**F**) Cells transfected with the plasmid to overexpress SURF4 were treated with 5 µM (**B**) or 10 µM (**C**) oAβ_1–42_ for 24 h. Data are the mean ± SEM of 4 independent experiments. ** *p* < 0.01 and * *p* < 0.5 vs. control by Student’s t-test.

**Figure 7 ijms-24-01278-f007:**
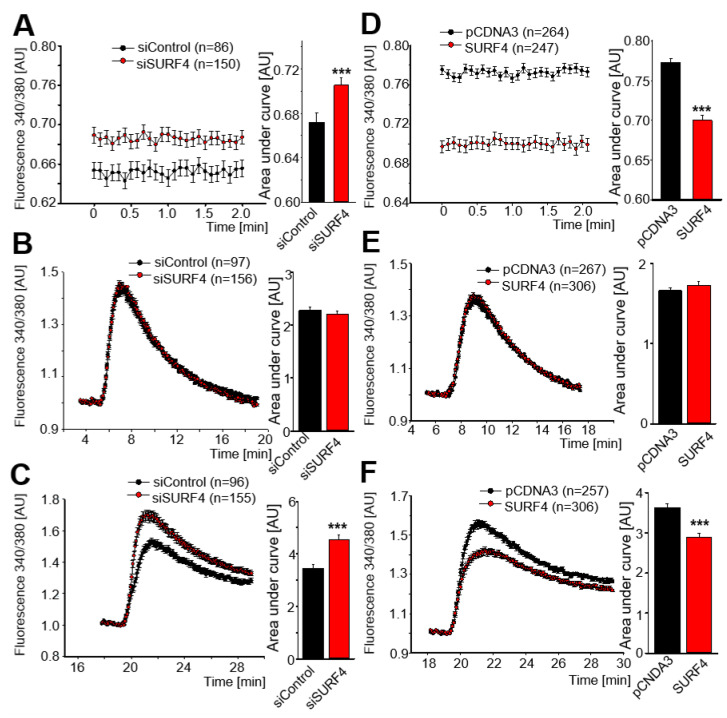
SURF4 conditions SOCE activity. (**A**,**B**) Human neuroblastoma cells were transfected with *SURF4* siRNA (**A**) or with a plasmid to overexpress SURF4 (**B**), and after 48 h the levels of calcium were measured by using FURA2. Data are the mean ± SEM of 4–7 independent experiments. *** *p* < 0.001 vs. control by Student’s t-test. (**C**,**D**) Cells transfected with *SURF4* siRNA (**C**) or SURF4 plasmid (**D**) were exposed to 0 extracellular calcium, and intracellular Ca^2+^ changes in response to ER depletion by thapsigargin were measured using FURA2. Data are the mean ± SEM of 4–6 independent experiments. (**E**,**F**) Cells were transfected with *SURF4* siRNA (**E**) or *SURF4* plasmid (**F**) during 48 h, and SOCE activity (induced by ER Ca^2+^ release with thapsigargin) was evaluated with FURA2 following re-addition of Ca^2+^ to the bathing solution. Data are the mean ± SEM of 5–7 independent experiments. *** *p* < 0.001 vs. control by Student’s t-test.

**Figure 8 ijms-24-01278-f008:**
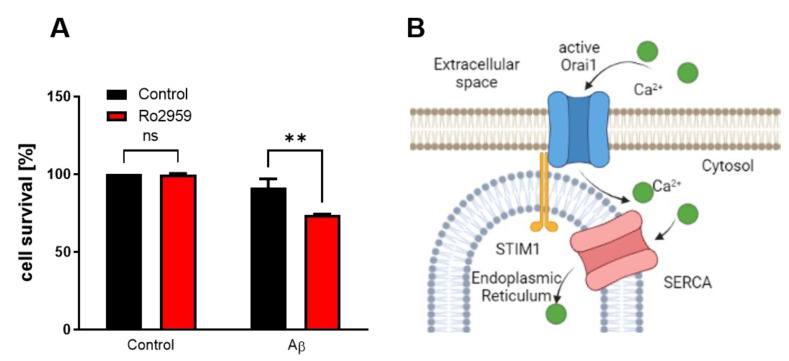
SURF4 affects SOCE proteins. (**A**) Neuroblastoma cells were treated with 10 µM oAβ_1–42_ in the presence of the SOCE inhibitor Ro for 24 h. Data are the mean ± SEM of 3 independent experiments. ** *p* < 0.01, non-significant (ns) vs. the respective controls by ANOVA plus Bonferroni as post-hoc test. (**B**) SOCE is dependent in the interaction of STIM, located in ER, with ORAI, located in the plasmatic membrane that opens to allow the entrance of calcium.

## Data Availability

Not applicable.

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
