# Peer review of "A Genome-Wide Functional Screen Identifies Enhancer and Protective Genes for Amyloid Beta-Peptide Toxicity"

_ijms, 2023, doi:10.3390/ijms24021278_

Round 1

Reviewer 1 Report

The paper entitled A Genome-wide Functional Screen Identifies Enhancer and Protective Genes for Amyloid Beta-Peptide Toxicity submitted by Pol Picón-Pagès et al. to the Int. J. Mol. Sci. is of high quality and very well presented. of course I am a little disappointed that the authors did not use multiparametric flow cytometry to follow the intracellular calcium level and the mitochondrial events that they described. But overall the present article could be accepted in its present form for publication.

Author Response

Dear Reviewer 1,

Thank you for the time you have spent reviewing our work.

Please find our response to your comments in the following paragraphs.

The authors.

“The paper entitled A Genome-wide Functional Screen Identifies Enhancer and Protective Genes for Amyloid Beta-Peptide Toxicity submitted by Pol Picón-Pagès et al. to the Int. J. Mol. Sci. is of high quality and very well presented. Of course, I am a little disappointed that the authors did not use multiparametric flow cytometry to follow the intracellular calcium level and the mitochondrial events that they described. But overall the present article could be accepted in its present form for publication..”

Response: Thank you very much for considering our work as of high quality and very well presented. We also appreciate very much your recommendation to use multiparametric flow cytometry to follow intracellular calcium and the mitochondrial events that we will apply in future works related with calcium homeostasis.

Reviewer 2 Report

The manuscript is devoted to the research of regulation of beta-amyloid toxicity. In the first part the authors show the confirming results of the used model adequacy. The second part is devoted to the search of regulatory genes for amyloid toxicity and to the explanation of the SURF4 enhancing effect by dysregulation of calcium homeostasis. The results of the research are original and can be recommended for publishing after minor revision of the presentation materials and the text:

1. in the introduction it is said that "i) they produce free radicals  (H2O2 and OH*)". But H2O2 is not a free radical but corresponds to reactive oxygen species;

2. confocal images (Fig. 1) were used to show punctuated distribution of beta-amyloid. But the images size don't allow to see it correctly;

3. Fig. 1E presents the result of cell growth rate of control cells and cells with WT, Artic and Dutch amyloids (4 bars). But in the X-axis name contains 5 elements, including GFP;

4. The parts of Fig. 6 description are mixed up. For example, Fig. 5D contains the results of determination of calcium level in the case of SURF4 overexpression. But in the Figure description these results are indicated by the letter B.

Author Response

Dear Reviewer 2,

Thank you for the time you have spent reviewing our work.

Please find our response to your comments in the following paragraphs.

The authors.

 “The manuscript is devoted to the research of regulation of beta-amyloid toxicity. In the first part the authors show the confirming results of the used model adequacy. The second part is devoted to the search of regulatory genes for amyloid toxicity and to the explanation of the SURF4 enhancing effect by dysregulation of calcium homeostasis. The results of the research are original and can be recommended for publishing after minor revision of the presentation materials and the text”

Response: Thank you for considering that our results are original, and it can be recommended for publication after revision.

Minor comments:

  1. In the introduction it is said that "i) they produce free radicals (H2O2 and OH·)". But H2O2 is not a free radical but corresponds to reactive oxygen species.

Response: the reviewer is completely right since H2O2 is not a free radical but a reactive oxygen species. We have modified the sentence and labelled in yellow as follows: “i) they produce reactive oxygen species (ROS), such as H2O2 and OH·,…”

  1. Confocal images (Fig. 1) were used to show punctuated distribution of beta-amyloid. But the images size doesn’t allow to see it correctly.

Response: Following referee’s recommendation we have enlarged the size of the confocal images to show the punctual distribution of the overexpressed Aß as you will find in the new Figure 1C.

  1. Fig. 1E presents the result of cell growth rate of control cells and cells with WT, Artic and Dutch amyloids (4 bars). But in the X-axis name contains 5 elements, including GFP.

Response: We apologize for the mistake. The conditions were Control, WT Aß, Arctic Aß and Dutch Aß. We have corrected the conditions in the X-axis of Figure 1E.

  1. The parts of Fig. 6 description are mixed up. For example, Fig. 6D contains the results of determination of calcium level in the case of SURF4 overexpression. But in the Figure description these results are indicated by the letter B.

Response: We apologize for the mistake. As the referee has noted the SURF4 overexpression comments are on Figure 6D and not B. We have corrected these items in the results section on Figure 6 and labelled in yellow.

Reviewer 3 Report

In this study, Picón-Pagès and colleagues performed a genome-wide yeast knockout screening analysis to investigate the Aβ1-42 toxicity. The authors identified various toxicity enhancers and protective genes. After, further gene ontology and pathway network analyses, one of the hits, SURF4 was selected to be further investigated. SURF4 overexpression was found to increase the Aβ1-42 toxicity, while its suppression decreased this response in neuroblastoma cells. SURF4 has previously proposed to serve as a repressor of calcium homeostasis mediator SOCE: In this study, chemical inhibition of SOCE resulted in increased toxicity in Aβ1-42 expressing cells compared to the wild type suggesting a potential SURF4 mediated SOCE activity-dependent mediation of Aβ1-42 toxicity.

The study has a significant and comprehensively investigated research plan. However, it suffers from some shortcomings including some controls. The authors are kindly asked to address the following major and minor points.

Major points:

1- Were expression levels of Aβ-GFP constructs normalized between the conditions in Fig.1?

2- (Section 2.1) - What could be the potential reason that authors previously observed toxicity of Aβ1-42 in yeast but others did not as clearly in the cytosol? Were expression levels comparable? (e.g. Can an increase in concentration drive their direction to secretory pathway components?). A discussion is suggested.

3- There are some subcellular compartmentalized clusters upon ectopic expression of  Aβ-GFP constructs in Fig.1C. Could authors please explain whether they are sole Aβ aggregates, or Aβ condensates comprising certain yeast proteins? Are they ThT-positive clusters?

4- Yeast genetic screening is consisting the backbone of the study, however, there is no main figure displaying the findings. Unfortunately, all the main outcome of this assay is solely represented in supplementary figures. I strongly suggest a new main figure (e.g. as a new Fig.1) showing a scheme of experimental approach (e.g. Fig.1a), then a volcano plot representation of toxicity level (Log fold-change as X-axis) versus significance (Log P-value as Y-axis) could be another subfigure. Ultimately, zooming into the “toxic” and “protective” gene populations from the volcano plot, and creating two heatmap representations of these populations including gene names will provide a great overall representation of the screening findings, and that would also help readers to better understand the experimental strategy and findings.

5- Authors have confirmed by low throughput qPCR and survival assays that one of the yeast knockout screening hits, SURF4 overexpression increased the cell toxicity. However, as the authors identified two gene populations from the yeast knockout screening data labeled as “toxic” and “protective”; it is necessary to provide a complete proof of concept verification: Authors are kindly asked to provide similar low throughput confirmation using a gene from the other fraction whose overexpression decreases the toxicity/increases the proliferation. 

Minor points:

1- There is no scale bar in Fig.1C

2- Text, axis labels, and legends in some figures are tiny and very hard to clearly read. The authors are suggested to make them larger, and the same size as possible throughout all the figures.

3- Typos in figure labels should be corrected (e.g. Aβ line above Fig.1C should not cover GFP. There is no GFP histogram bar but a GFP label in Fig.1E).

Round 2

Reviewer 3 Report

In the revised version, the authors significantly improved the manuscript by addressing my questions and suggestions.